# Association of Circulating Markers of Microbial Translocation and Hepatic Inflammation with Liver Injury in Patients with Type 2 Diabetes

**DOI:** 10.3390/biomedicines12061227

**Published:** 2024-05-31

**Authors:** Leila Gobejishvili, Vatsalya Vatsalya, Diana V. Avila, Yana B. Feygin, Craig J. McClain, Sriprakash Mokshagundam, Shirish Barve

**Affiliations:** 1Department of Physiology, University of Louisville School of Medicine, Louisville, KY 40202, USA; 2Division of Gastroenterology, Hepatology and Nutrition, Department of Medicine, University of Louisville School of Medicine, Louisville, KY 40202, USA; vatsalya.vatsalya@louisville.edu (V.V.); craig.mcclain@louisville.edu (C.J.M.); 3Department of Pharmacology and Toxicology, University of Louisville School of Medicine, Louisville, KY 40202, USA; davila@houstonmethodist.org; 4Data Science Core, Norton Research Institute, Department of Pediatrics, University of Louisville School of Medicine, Louisville, KY 40202, USA; yana.feygin@louisville.edu; 5Robley Rex Veterans Affairs Medical Center, Louisville, KY 40206, USA; sriprakash.mokshagundam@louisville.edu; 6Division of Endocrinology, Metabolism & Diabetes, Hepatology and Nutrition, Department of Medicine, University of Louisville School of Medicine, Louisville, KY 40202, USA

**Keywords:** endotoxemia, sCD14, sCD163, liver injury, T2DM

## Abstract

Background: Virtually the entire spectrum of liver disease is observed in association with type 2 diabetes mellitus (T2DM); indeed, T2DM is now the most common cause of liver disease in the U.S. We conducted a pilot study to investigate the relevance of increased microbial translocation and systemic inflammation in the development of liver injury in patients with T2DM. Methods: Patients with T2DM (n = 17) and non-diabetic controls (NDC; n = 11) aged 25–80 yrs. participated in this study. Serum levels of endotoxin, calprotectin, soluble CD14 and CD163, and several inflammatory cytokines were measured. In addition to standard liver injury markers, ALT and AST, novel serum markers of liver injury, keratin 18 (K-18) M30 (apoptosis-associated caspase-cleaved keratin 18), and M65 (soluble keratin 18) were evaluated. Statistical analyses were performed using the Mann–Whitney test to assess differences between study groups. Pearson’s correlation analysis was performed to determine the strength of association between two variables using GraphPad Prism 9.5.0 software. Results: Patients with T2DM had significantly higher levels of sCD14 in comparison to NDC, suggesting an increase in gut permeability, microbial translocation, and monocyte/macrophage activation. Importantly, relevant to the ensuing inflammatory responses, the increase in sCD14 in patients with T2DM was accompanied by a significant increase in sCD163, a marker of hepatic Kupffer cell activation and inflammation. Further, a positive correlation was observed between sCD163 and endotoxin and sCD14 in T2DM patients but not in NDC. In association with these changes, keratin 18 (K-18)-based serum markers (M65 and M30) that reflect hepatocyte death were significantly higher in the T2DM group indicating ongoing liver injury. Notably, both M65 and M30 levels correlated with sCD14 and sCD163, suggesting that immune cell activation and hepatic inflammation may be linked to the development of liver injury in T2DM. Conclusions: These findings suggest that the pathogenic changes in the gut–liver axis, marked by increased microbial translocation, may be a major component in the etiology of hepatocyte inflammation and injury in patients with T2DM. However, larger longitudinal studies, including histological evidence, are needed to confirm these observations.

## 1. Introduction

Diabetes mellitus (DM) affects approximately 38 million Americans (https://diabetes.org, accessed on 1 April 2024) and over 422 million individuals worldwide (www.who.int/diabetes/facts/world_figures/en/, accessed on 1 April 2024). More than 90% of individuals with diabetes have type 2 diabetes (T2DM) (https://www.cdc.gov/diabetes/basics/type2.html, accessed on 1 April 2024), and it is considered the most common form of diabetes. It has been reported that diabetes is the direct cause of 1.5 million deaths each year. T2DM is a heterogeneous disease with varying degrees of severity, disease progression, response to therapeutic interventions, and complications [1]. While much attention has been given to the classic micro- and macro-vascular complications of diabetes, there is increasing recognition of the role of liver abnormalities in diabetes-related outcomes [2,3]. Patients with Type 2 diabetes frequently have liver disease ranging from metabolic dysfunction-associated steatotic liver disease (MASLD) and metabolic dysfunction-associated steatohepatitis (MASH) to cirrhosis and hepatocellular carcinoma [4,5,6,7,8,9,10]. In this regard, diabetes has been shown to be a risk factor for the development of liver disease and to accelerate its progression to more severe forms of liver disease [3,4,6,11]. Moreover, the diagnosis of liver disease in T2DM patients is associated with the increased incidence and prevalence of both macrovascular and microvascular complications of T2DM [12,13,14], risk of death, and other associated complications [15,16].

It has been postulated that an altered gut microbiome may be a key factor in the development of T2DM [17]. Studies in animal models of obesity and T2DM have shown a significant association between alterations in the gut microbiome and obesity, insulin resistance, and diabetes mellitus [18,19,20,21,22]. Qualitative and quantitative alterations in gut microbiota (dysbiosis) have also been noted in patients with T2DM [23,24]. Translocation of bacterial products and bacteria occurring due to the alterations in the intestinal microflora and intestinal epithelial barrier dysfunction have been demonstrated to contribute to the systemic endotoxemia, immune activation, and inflammation seen in patients with diabetes [23,24,25,26,27]. Indeed, chronic low-grade systemic inflammation has been identified as a risk factor for the development of diabetes and its complications [25,28,29,30]. In this regard, it has been shown that patients with T2DM have increased numbers of activated monocytes (CD14^+^CD16^+^-positive) with increased expression of pro-inflammatory cytokines and chemokines, including IL-1β, TNF, IL-6, COX-2, IL-8, ICAM-1 [31,32].

We hypothesized that enhanced gut-barrier dysfunction and the resultant systemic microbial translocation-mediated inflammation would be a significant feature of T2D-related liver disease pathogenesis. Accordingly, we analyzed the role of systemic indicators of microbial translocation in immune activation/inflammation and its association with liver injury markers in patients with T2DM. Our results show that patients with T2DM have significantly higher levels of systemic markers of gut microbial translocation and macrophage activation, sCD14 and sCD163. Moreover, we show that these patients have elevated levels of K-18 M65 and M30. Notably, sCD14 and sCD163 levels strongly correlate with both M30 and M65 in patients with T2DM but not in non-diabetic controls, showing the contributory role of microbial translocation and immune cell activation to liver injury.

## 2. Materials and Methods

### 2.1. Study Design and Participants

Inclusion Criteria: “Subjects” included those with T2DM who were between the ages of 18 to 80 years, with BMI < 50, and who were not taking thiazolidenedione (Rosiglitazone or Pioglitazone). “Controls” included individuals without obesity (BMI 18–29.9) who were otherwise healthy. Exclusion criteria included for subjects with diabetes mellitus also apply to controls. The following exclusion criteria were used: (1) Being a pregnant or lactating female; (2) Smoking; (3) Taking steroid medication; except for topical steroids; (4) Taking systemic antibiotics within one month of baseline; (5) Taking thiazolidenedione (Rosiglitazone or pioglitazone); (6) Presence of diabetes related wounds and/or ulcers; (7) Having taken an investigational drug within 30 days of enrollment; (8) Presence of other significant infections; and/or (9) Currently having, or having had a history of (within three months), the following diseases: Severe cardiovascular disease, Severe pulmonary disease, Severe liver disease, Active malignancy, Cerebral vascular disease, HIV, TB, hepatitis or other active infectious disease, Drug or alcohol abuse, Mental or cognitive deficiencies, End stage renal disease as indicated by current dialysis treatment, Renal insufficiency as indicated by physician report/records indicating serum creatinine level of 3.0 mg/dl or greater, Psychosis, MI or CABG in the previous 6 months.

Participants were recruited under a study approved by the Institutional Review Board of the University of Louisville. All study participants were informed about the purpose of the study and any potential risks/side effects. Informed consent was obtained from all Subjects and Controls before participation in the study. Eleven healthy, non-diabetic controls (NDC—2 males and 9 females) and 17 patients with type 2 diabetes (T2DM-9 males and 8 females) were enrolled in this study. Whole blood was drawn at fasting, centrifuged, and serum was aliquoted and stored at −80 °C for future analysis. Age and BMI for all participants and fasting HgBA1C, glucose, and insulin levels for T2DM patients were recorded. The patients with T2DM were on anti-diabetic medications metformin (n = 11), sulfonylurea (n = 4), and/or insulin (n = 12).

### 2.2. Serum Soluble CD14 (sCD14), CD163 (sCD163) and Endotoxin

Serum sCD14 levels were measured by using a human sCD14 ELISA kit (Hycult Biotech, Uden, The Netherlands) according to the manufacturer’s instructions. The serum was diluted 100 times with the supplied dilution buffer before performing the assay. Serum sCD163 levels were measured using a human CD163 Quantikine ELISA kit (R&D Systems, Inc., Minneapolis, MN, USA). The serum was diluted 10 times with the supplied calibrator diluent. Serum endotoxin levels were measured using a kinetic QCL-Limulus Amoebocyte Lysate assay (Lonza, Walkersville, MD, USA).

### 2.3. Serum K18M30 and M65 Levels

Serum M30 (apoptosis-associated caspase-cleaved keratin 18) and M65 (soluble keratin 18) levels were quantified by using M30 and M65 ELISA kits (diaPharma, Columbus, OH, USA) according to the manufacturer’s instructions.

### 2.4. Serum Cytokines, Chemokines, and Hormones

Serum cytokines, chemokines, and hormones were quantified by multi-analyte, milliplex human cytokine/chemokine and adipokine kits (Milliplex Cytokine/Chemokine and Adipokine magnetic panel, EMD Millipore, Billerica, MA, USA) on the Luminex (Luminex, Austin, TX, USA) platform as previously described [33]. Specifically, interleukin-6 (IL-6), interleukin-8 (IL-8), monocyte chemotactic protein-1 (MCP-1), plasminogen activator inhibitor-1 (PAI-1), tumor necrosis factor (TNFα), adiponectin, and leptin were measured.

### 2.5. Statistical Analysis

Statistical analyses were performed using GraphPad Prism version 9.5.0 for Windows (GraphPad Software Inc., La Jolla, CA, USA). Differences between study groups were analyzed by using the non-parametric unpaired Mann–Whitney test. Individual values for each group are presented as mean ± SD. A *p* ≤ 0.05 was considered statistically significant. Pearson correlation analysis was performed to evaluate associations between the variables measured. Pearson r and P (two-tailed) values are indicated for significant correlations only. A post hoc power analysis indicated the available study sample was sufficient for 80% power to detect significant Pearson correlations of at least 0.63 and standardized mean differences (Cohen’s d) greater than or equal to 1.2 for the Mann–Whitney U test.

## 3. Results

### 3.1. The Characteristics of Study Participants

There was a significant difference in age and BMI between T2DM and non-diabetic control (NDC) participants (Table 1). The age range for the T2DM patients was 42–80 yrs. and for the control group was 25–58 yrs. Patients with T2DM had obesity by BMI assessments compared to the non-diabetic controls. Therefore, we used age and BMI as factors in analysis wherever applicable. Although T2DM patients were on anti-diabetic medication, glucose levels were higher in those patients than in the controls (Table 1). HbA1C levels were between 6.5 and 14.3 in the T2DM group (Table 1). High glucose and A1c levels in patients with T2DM indicate that this patient population was poorly controlled. All patients with T2DM had hyperlipidemia. AST and albumin levels did not differ between the groups; however, total protein was lower and alkaline phosphatase levels higher in patients with T2DM, indicating some degree of liver damage (Table 1). Notably, although ALT levels were significantly higher in the T2DM group, the levels were in the control range.

### 3.2. Gut Microbial Translocation and Immune Activation Markers in Patients with T2DM

Plasma endotoxin and calprotectin levels were not significantly different among the groups (Figure 1A,B). However, the levels of sCD14, a more stable marker of previous endotoxin exposure [34], were significantly elevated in T2DM patients in comparison to non-diabetic controls (Figure 1C), indicating an increase in gut-derived microbial translocation. Importantly, the increase in sCD14 in T2DM patients was accompanied by a significant increase in sCD163 (Figure 1D), a soluble marker of macrophage activation and systemic inflammation [35,36,37,38].

To assess the functional relation between microbial translocation and immune activation, we performed a Pearson correlation analysis between endotoxin and sCD14 and sCD163 levels in both patients with T2DM and non-diabetic controls (NDC). Interestingly, we did not find a significant correlation between endotoxin and sCD14; however, the correlation between endotoxin and sCD163 was strong with r = 0.4792 but did not reach significance (*p* = 0.0516) in patients with T2DM but not in NDCs. Further, we found a strong and significant correlation between sCD14 and sCD163 levels in patients with T2DM (r = 0.5798; *p* = 0.0186) but not in NDCs (Figure 2).

### 3.3. Serum Levels of Inflammatory Cytokines, Chemokines and Hormones

We next examined serum levels of several inflammatory cytokines and chemokines in non-diabetic control (NDC) and T2DM groups. IL-6, IL-8, TNF, MCP-1, and PAI-1 levels trended higher in the T2DM group; however, they did not reach significance (Figure 3A). Serum levels of leptin, resistin, and insulin tended to be higher, while adiponectin levels were lower in patients with T2DM (Figure 3B).

### 3.4. Liver Injury Markers—M30 and M65

Serum K-18 has been proposed to be a more accurate assessment of liver injury of different etiologies. Hence, we measured serum K-18 levels, both M30 (apoptosis-associated caspase-cleaved keratin 18) and M65 (soluble keratin 18). Serum M30 and M65 levels were in a normal range in NDC study subjects. In comparison, T2DM patients had higher M65 and M30 levels, indicating ongoing hepatocyte apoptosis and necrosis (Figure 4). Notably, one individual in the NDC group showed very high levels of M30, which resulted in statistically non-significant differences between the groups.

### 3.5. Correlation of Serum Markers of Microbial Translocation, Macrophage Activation, and Liver Injury in Patients with T2DM

Since gut-derived microbial translocation and immune activation have been shown to play a critical role in the development of liver injury [39,40,41], we further examined if there was a correlation between the markers of liver injury and immune activation with bacterial translocation. Indeed, our analysis showed a significant moderate to strong correlation of M65 with sCD14 (Figure 5A) and M65 and sCD163 (Figure 5B) in patients with T2DM but not in the NDC group. Moreover, M30 levels were strongly correlated with sCD14 and sCD163 in the T2DM group but had no significant correlation in NDCs (Figure 6A,B). Taken together, these results indicate a strong connection between microbial translocation, immune cell activation, and the development of liver injury in patients with T2DM. Notably, a strong correlation between sCD163 and M30 levels indicates a role for macrophage activation in apoptotic liver injury in patients with T2DM.

## 4. Discussion

Type 2 diabetes mellitus (T2DM), a chronic metabolic disorder, is marked by the presence and development of multiple complications including liver disease [42,43,44,45,46]. Almost the entire spectrum of liver disease is observed in association with T2DM, including abnormal liver enzymes, metabolic dysfunction-associated steatotic liver disease (MASLD), cirrhosis, hepatocellular carcinoma, and acute liver failure. Indeed, T2D has been shown to be a risk factor for the development of liver disease and to accelerate its progression to more severe forms of liver disease [3,4,6,11]. Increased adiposity in T2DM is associated with the dysregulated release of adipokines, adiponectin, resistin, and leptin by white adipose tissue (WAT) [47,48]. Importantly, WAT dysfunction and increased lipolysis due to insulin resistance leads to excess free fatty acid accumulation in the liver, which can cause mitochondrial dysfunction and endoplasmic reticulum stress in hepatocytes and liver injury [49]. Although the prevalence of liver disease is well established in T2DM, understanding the underlying pathogenic mechanisms is not fully developed. Intestinal barrier dysfunction and ensuing gut microbial translocation and systemic inflammation are essential components of the pathogenic changes in the “gut–liver axis” leading to liver disease. In this regard, alterations in gut microbiota have been documented in subjects with T2DM [50,51,52,53]; however, the understanding of the relevance of gut-derived events leading to hepatic inflammation and injury as a diabetic complication remains to be fully clarified. Hence, in the present pilot study, we examined the association of gut permeability and microbial translocation markers with markers of systemic/hepatic inflammation and liver injury.

Increased disturbance in the “gut–liver axis” was clearly indicated in patients with T2D by a significant increase in sCD14 levels, an indirect marker of increased gut permeability and/or mucosal damage and microbial translocation [54]. The increased plasma levels of sCD14 predominantly depict responses to bacterial endotoxin-mediated activation of peripheral blood monocytes and are associated with metabolic endotoxemia linked to metabolic syndrome, obesity, and insulin resistance [27,55,56]. Although there was a significant increase in sCD14 in patients with T2D compared to non-diabetic controls (NDC), a corresponding increase in endotoxin (LPS) was not observed. This discrepancy could potentially be due to the formation of sCD14-LPS complexes, thereby rendering LPS inaccessible for detection to measure changes in endotoxin levels [56].

With regard to the present pilot study, earlier work has clearly demonstrated the presence of gut-derived endotoxemia and its role in hepatic inflammation and injury in the different forms of human liver disease, including nutritional deficiency and toxicant-induced liver injury [57]. Hence, to investigate the presence of liver disease in patients with T2DM, the pathogenic relevance of sCD14, an indicator of gut-derived microbial translocation and endotoxemia, was assessed in the context of markers of hepatic inflammation and injury in patients with T2DM. The liver is exposed to gut-derived endotoxin via the portal vein circulation, leading to the activation of the hepatic resident macrophages—Kupffer cells. Activation of hepatic Kupffer cells is associated with the production and excretion of soluble CD163 [58,59]. The presence of plasma sCD163 has been identified as a key marker of Kupffer cell activation and hepatic inflammation and is highly associated with liver injury and fibrosis [36]. Indeed, sCD163 has been shown to correlate with systemic inflammation and liver injury of different etiologies [35,36,37,38]. Importantly, a robust and significant correlation between sCD163 and sCD14 as well as between sCD163 and endotoxin was observed in patients with T2D but not in non-diabetic controls. These data indicated that the pathogenic changes in the gut–liver axis leading to an increase in gut-derived endotoxemia was linked with hepatic Kupffer cell activation and inflammation observed in the T2D patients.

Examination of keratin 18 (K-18)-based serum markers (M65 and M30) that reflect hepatocyte death [60,61] demonstrated that along with gut-driven hepatic inflammation, there was a significant liver injury in patients with T2D. Hepatocyte death is a significant pathogenic component in the onset and development of liver injury and an indicator of the severity of liver disease, including fibrosis and cirrhosis [62,63]. There are several studies that have examined serum-based K-18 M30 and M65 as markers of hepatocyte death in many forms of liver disease, including MASLD, MASH, and alcohol-associated liver disease (ALD) [64,65]. However, there are very few studies that have examined the levels of serum K-18 to detect the presence of liver injury in T2DM [10,66,67] or that have investigated its association with gut-derived endotoxemia and hepatic inflammation. Notably, a significant relationship between K-18 hepatocyte death markers and markers of gut-derived pathogenic events (i.e., microbial translocation/endotoxemia (sCD14) and hepatic inflammation/Kupffer cell activation (sCD163) was observed. These findings strongly support the notion that the pathologic changes in the gut–liver axis may be a major component in the etiology of hepatocyte inflammation and injury and development of MASLD/MASH, the most common form of liver disease in people with T2DM.

This study has several limitations. It is a pilot study with small numbers of participants which limited the scope of the analysis. The control group was not age-, sex-, or BMI-matched to the patients. Additionally, patient records did not have information regarding certain medications on all study subjects. Despite these limitations, interesting insights were gained concerning gut–liver dysfunction in T2DM, which warrant further studies. Larger longitudinal studies, including histological evidence, are needed to confirm these observations while adjusting for important confounders such as patient age, sex, and BMI.

## 5. Conclusions

In summary, data from this pilot study provide important information about different aspects of gut–liver pathology that can be useful in diagnosing liver diseases, assessing disease severity, monitoring disease progression, and evaluating treatment response in patients with T2DM. Integrating these gut–liver pathogenic markers into clinical practice could help bring a better understanding of the underlying mechanisms of liver injury in diabetes and develop tailored treatment strategies accordingly.

## Figures and Tables

**Figure 1 biomedicines-12-01227-f001:**
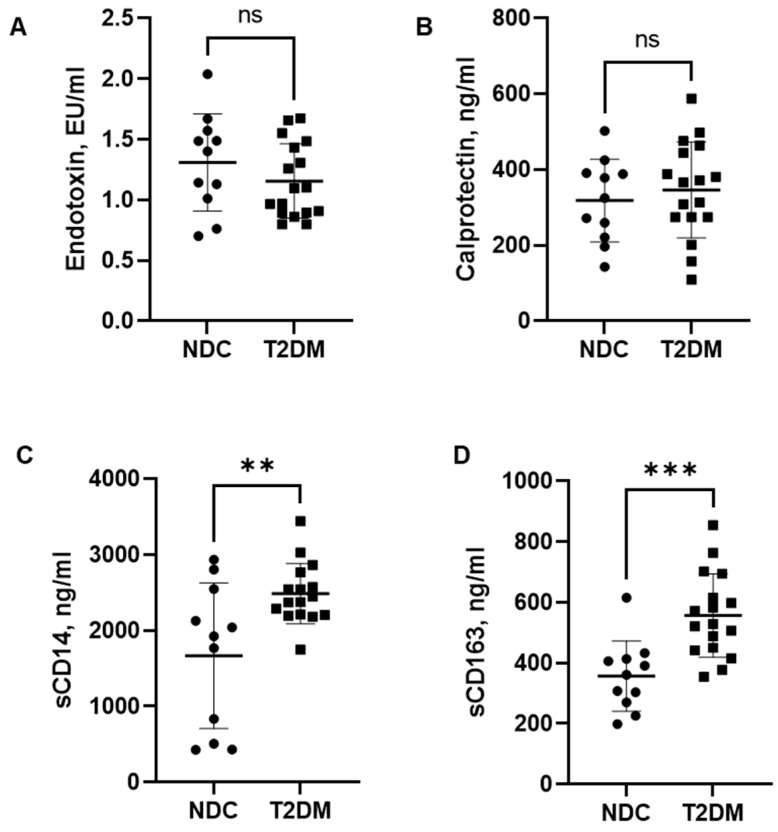
Serum markers of gut microbial translocation. (**A**) Endotoxin levels, (**B**) Calprotectin, (**C**) Soluble CD14 levels, (**D**) Soluble sCD163. Data are presented as individual values with mean ± SD. Mann–Whitney test. ** *p* < 0.01, *** *p* < 0.001, ns—non-significant.

**Figure 2 biomedicines-12-01227-f002:**
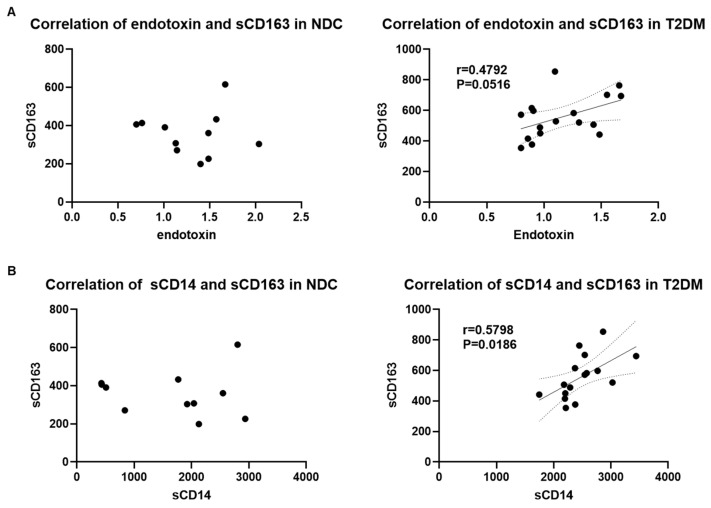
Elevated serum soluble CD163 levels correlate with endotoxin and sCD14 levels in patients with T2DM. (**A**) Pearson correlation analysis between endotoxin and sCD163 levels in NDC and T2DM groups, (**B**) Pearson correlation analysis of sCD14 and sCD163 levels in NDC and T2DM groups. P and r values are indicated.

**Figure 3 biomedicines-12-01227-f003:**
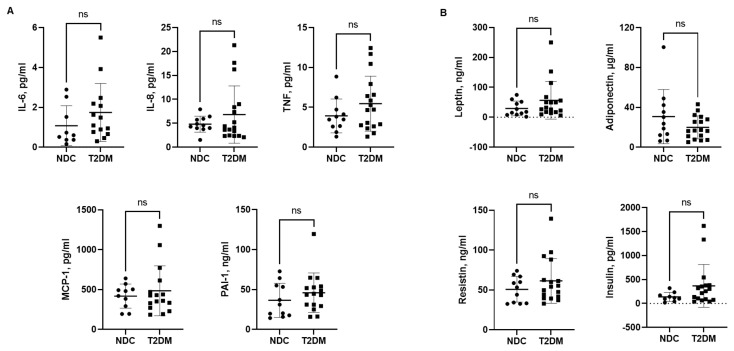
Serum levels of cytokine, chemokine, and hormones in patients with T2DM and non-diabetic controls. (**A**) IL-6, IL-8, TNF, MCP-1, and PAI-1 levels, (**B**) Leptin, adiponectin, resistin, and insulin levels. Data are presented as individual values with mean ± SD, ns -non-significant.

**Figure 4 biomedicines-12-01227-f004:**
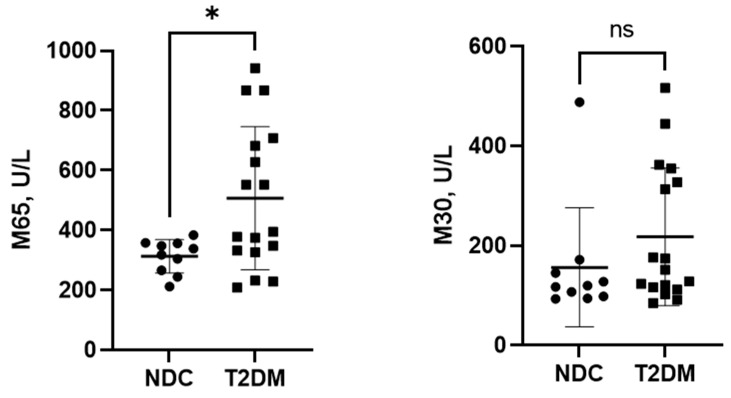
Serum levels of liver injury markers M65 and M30. Data are presented as individual values with mean ± SD, * *p* < 0.05, ns—non-significant.

**Figure 5 biomedicines-12-01227-f005:**
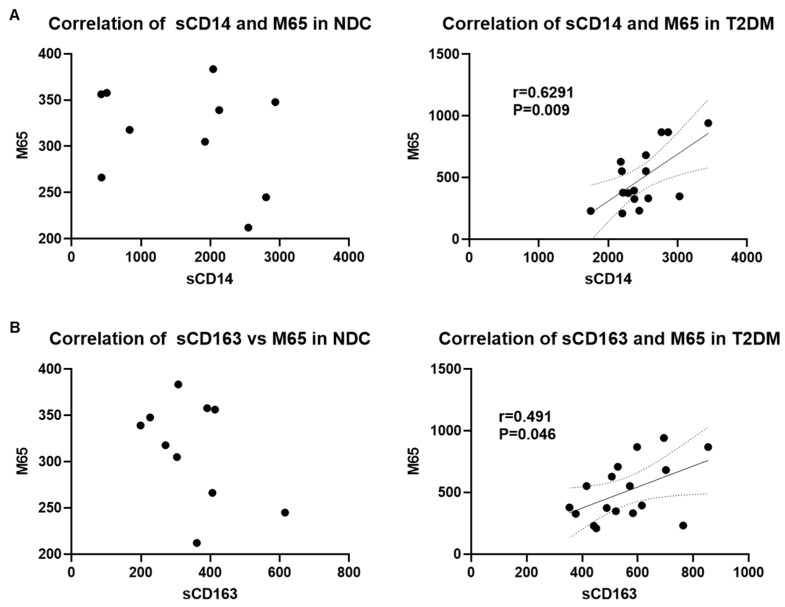
The liver injury marker, M65, correlates with sCD14 and sCD163 in patients with T2DM. (**A**) Pearson correlation analysis of serum sCD14 and M65 levels in NDC and T2DM groups, (**B**) Pearson correlation analysis of serum sCD163 and M65 levels in NDC and T2DM groups. P and r values are indicated.

**Figure 6 biomedicines-12-01227-f006:**
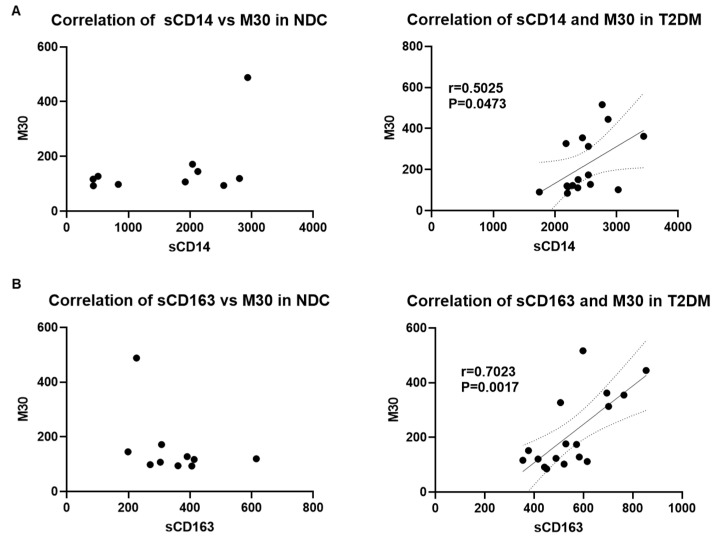
The liver injury marker, M30, correlates with sCD14 and sCD163 in patients with T2DM. (**A**) Pearson correlation analysis of serum sCD14 and M30 levels in NDC and T2DM groups, (**B**) Pearson correlation analysis of serum sCD163 and M30 levels in NDC and T2DM groups. P and r values are indicated.

**Table 1 biomedicines-12-01227-t001:** Baseline characteristics of the healthy control and diabetic participants.

Variables	T2DM (n = 17)	NDC (n = 11)	*p*-Value
Age, yrs.	61.1 ± 10.4	39.9 ± 12.6	≤0.001
Sex	8 females, 9 males	9 females, 2 males	NA
BMI	36.1 ± 7.1	23.9 ± 3.1	≤0.001
HgbA1c	9.1 ± 1.9	NA	NA
ALT	19.1 ± 8.03	15.3 ± 15.33	=0.031
AST	20.4 ± 6.67	17.6 ± 4.38	=NS
Total Protein	7.4 ± 0.45	7.8 ± 0.46	=0.054
Albumin	4.9 ± 0.26	5 ± 0.32	NS
Alkaline Phosphatase	88.8 ± 23.37	62.1 ± 12.33	=0.0029
Glucose (mg/dL)	187.4 ± 57.2	85.6 ± 3.9(5)	=0.001

Data are presented as mean ± SD. Mann–Whitney U-test was performed, and *p* values are indicated to show the significance of the difference between patients with T2DM and non-diabetic, healthy controls (NDC). NS = non-significant, NA = not applicable.

## Data Availability

The data presented in this study are available upon request from the corresponding author.

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
