# Peer review of "Association of Circulating Markers of Microbial Translocation and Hepatic Inflammation with Liver Injury in Patients with Type 2 Diabetes"

_biomedicines, 2024, doi:10.3390/biomedicines12061227_

Round 1

Reviewer 1 Report

Comments and Suggestions for Authors

This study reported the findings from a pilot study to determine the association of circulating markers of microbial translocation and hepatic inflammation with liver injury in patients with T2DM.  The investigators recruited 17 diabetic patients and 11 nondiabetic patients who provided fasting serum for assays of glucose, HbA1c, insulin and many biomarkers.  The differences of these metabolic parameters and biomarkers between diabetic and nondiabetic patients were analyzed.  Although the findings are interesting, there are several weaknesses in this study that need to be addressed to improve the manuscript:

1. The study was focused on the association of circulating markers of microbial translocation and hepatic inflammation with liver injury in patients with T2DM.  However, it seems that the patients recruited in this study do not have T2DM-associated nonalcoholic steatohepatitis (NASH) which is characterized by hepatic inflammation and liver injury.  The authors need to address this issue.

2. The sample sizes in this study are small (17 and 11 for diabetic and nondiabetic patients, respectively) given the large variations in the metabolic parameters and markers in patients.  Thus, power analysis is important to determine if the small sample sizes are okay for the correlation analysis.  

3. The average age and BMI are significantly different between diabetic and nondiabetic groups (age: 61 vs 40, p<0.001; BMI, 36 vs 24, p<0.001).  It is unclear why the study did not match the age and BMI of nondiabetic patients with those of diabetic patients.  The authors need to address this issue.

4. In addition to hyperglycemia, dyslipidemia also plays an important role in microbial translocation and hepatic inflammation with liver injury in patients with T2DM.  Thus, the patient clinical information (Table 1) should include blood lipids such as cholesterol, triglycerides, free fatty acids.

5. The patient clinical information should also include the medications that potentially affect systemic and liver inflammation such as aspirin, statins and omega-3 fish oil.

6. In Fig. 4, it is not clear if serum M65 in diabetic patients is significantly higher than that in nondiabetic patients since the difference between diabetic and nondiabetic patients is indicated by symbol * but only ** is <0.05.

Author Response

We thank the reviewers for their thoughtful comments and appreciate their constructive suggestions. All the appropriate changes are highlighted in yellow in the revised manuscript. We hope the revised version of the manuscript, which systematically addresses the comments made by the reviewers, will be acceptable for publication.

We have also performed English language editing (not marked) and replaced Figure 2, since one of the panels on the figure had incorrect r value (5798 instead of 0.5798).

Following is the detailed response to Reviewer’s comments:

Reviewer 1:

This study reported the findings from a pilot study to determine the association of circulating markers of microbial translocation and hepatic inflammation with liver injury in patients with T2DM.  The investigators recruited 17 diabetic patients and 11 nondiabetic patients who provided fasting serum for assays of glucose, HbA1c, insulin and many biomarkers.  The differences of these metabolic parameters and biomarkers between diabetic and nondiabetic patients were analyzed.  Although the findings are interesting, there are several weaknesses in this study that need to be addressed to improve the manuscript:

  1. The study was focused on the association of circulating markers of microbial translocation and hepatic inflammation with liver injury in patients with T2DM.  However, it seems that the patients recruited in this study do not have T2DM-associated nonalcoholic steatohepatitis (NASH) which is characterized by hepatic inflammation and liver injury.  The authors need to address this issue.

Response: We purposely selected patients with T2DM who did not have standard biochemical evidence of MASH (e.g. high liver injury markers ALT/AST). The point of this investigation was to determine whether patients with T2DM without MASH had alterations in gut barrier function and presented evidence of early liver disease. Accordingly, we assessed serum liver injury markers, cytokeratin 18 (CK-18) M30 and M65, that have been demonstrated to be more sensitive and accurate measures of liver injury of different etiologies.

  1. The sample sizes in this study are small (17 and 11 for diabetic and nondiabetic patients, respectively) given the large variations in the metabolic parameters and markers in patients.  Thus, power analysis is important to determine if the small sample sizes are okay for the correlation analysis.  

Response: We thank the reviewer for the comment and apologize for the lapse in including the power analysis.  A post-hoc power analysis was indeed performed which indicated that the available study sample was sufficient for 80% power to detect significant Pearson correlations of at least 0.63 and standardized mean differences (Cohen’s d) greater than or equal to 1.2 for the Mann-Whitney U test. We have appropriately included the power analysis in the “Methods Section” in the revised manuscript.

  1. The average age and BMI are significantly different between diabetic and nondiabetic groups (age: 61 vs 40, p<0.001; BMI, 36 vs 24, p<0.001).  It is unclear why the study did not match the age and BMI of nondiabetic patients with those of diabetic patients.  The authors need to address this issue.

Response: We agree with the reviewer that the average age and BMI are significantly different between diabetic and nondiabetic groups. However, since the goal of this pilot study was to evaluate the presence of liver injury in the context of Type 2 diabetes, nondiabetic individuals without liver disease and BMIs ranging from 19 to 29 were recruited as study controls. We have appropriately addressed this issue in the revised manuscript as a study limitation.   

  1. In addition to hyperglycemia, dyslipidemia also plays an important role in microbial translocation and hepatic inflammation with liver injury in patients with T2DM.  Thus, the patient clinical information (Table 1) should include blood lipids such as cholesterol, triglycerides, free fatty acids.

Response: All the participating T2DM patients had hyperlipidemia and was noted in their clinical charts. However, unfortunately due to logistical limitations we could not acquire specific values for blood lipids for all the study subjects. We have appropriately revised “The Characteristics of Study Participants” section to reflect the hyperlipidemic status.    

  1. The patient clinical information should also include the medications that potentially affect systemic and liver inflammation such as aspirin, statins and omega-3 fish oil.

Response: We agree with the comment, however information regarding medications such as aspirin, statins and omega-3 fish oil was not available in the patient medical records.

  1. In Fig. 4, it is not clear if serum M65 in diabetic patients is significantly higher than that in nondiabetic patients since the difference between diabetic and nondiabetic patients is indicated by symbol * but only ** is <0.05.

Response: M65 levels were indeed significantly higher in patients with T2DM. However, ** indicating statistical significance was a typographical error and has been rectified to *P<0.05 in the revised manuscript (Figure 4 legend).

Reviewer 2 Report

Comments and Suggestions for Authors

The submitted manuscript is interesting, well designed, well written and easy to read. The research topic is relatively novel, especially with the markers the researchers used in their research. The methodology is not very novel, however it is sufficient to answer the research question posed.

However, I have a few observations.

1. The sample size of the groups used is very small and this would possibly be the cause of observing only trends in most of the analyses. It also prevents a deeper analysis, e.g. adjusting for age and sex.

2. The authors mention that there is a strong correlation between endotoxin and sCD163, and although the r is significant, the p is not significant. Figure 2A. 

Author Response

We thank the reviewers for their thoughtful comments and appreciate their constructive suggestions. All the appropriate changes are highlighted in yellow in the revised manuscript. We hope the revised version of the manuscript, which systematically addresses the comments made by the reviewers, will be acceptable for publication.

We have also performed English language editing (not marked) and replaced Figure 2, since one of the panels on the figure had incorrect r value (5798 instead of 0.5798).

Following is the detailed response to Reviewer’s comments:

Reviewer 2:

The submitted manuscript is interesting, well designed, well written and easy to read. The research topic is relatively novel, especially with the markers the researchers used in their research. The methodology is not very novel, however it is sufficient to answer the research question posed.

However, I have a few observations.

  1. The sample size of the groups used is very small and this would possibly be the cause of observing only trends in most of the analyses. It also prevents a deeper analysis, e.g. adjusting for age and sex.

Response: We agree with the reviewer. We added a limitation section addressing the sample size in our revised manuscript.

  1. The authors mention that there is a strong correlation between endotoxin and sCD163, and although the r is significant, the p is not significant. Figure 2A. 

Response: We agree with the reviewer and have appropriately revised the manuscript indicating that the correlation between endotoxin and sCD163 did not reach significance (P=0.0516).

Round 2

Reviewer 1 Report

Comments and Suggestions for Authors

The authors have addressed the concerns from the reviewer and revised the manuscript. The manuscript is improved.

Reviewer 2 Report

Comments and Suggestions for Authors

The investigators addressed my observations appropriately.